# Relationship between physical activity and neighborhood environment in preschool children during COVID-19.—A cross-sectional study using 24-hour activity records-

NAOKI YAMAGUCHI[1,2]*, TAKAHIKO FUKUMOTO[1], HIDETAKA IMAGITA[3]

**1** Graduate School of Health Sciences, Kio University, Nara, Japan, **2** Department of Rehabilitation, Kiwa Hospital, Wakayama, Japan, **3** Faculty of Health Science and Welfare, Saitama Prefectural University, Saitama, Japan

\* f0985561@kio.ac.jp

**Data Availability Statement:** All relevant data are within the manuscript.

## Abstract

Maintaining physical activity and the development of physical and cognitive functions are essential especially in vulnerable populations such as children. Considering the substantial impact of the COVID-19 epidemic on preschool children and the lack of research focusing on this age group, this study examined the duration of physical activity among preschool-aged children on weekend days and its relationship with the environment. This study was conducted between October and November 2021, with the cooperation of 357 parents of preschoolers attending kindergartens, nurseries, and childcare centers. Physical activity was assessed using a 24h physical activity record. A questionnaire was used to collect basic data on the neighborhood environment. Stepwise multiple regression analysis was used to analyze the relationship between the five physical activity measures (sedentary time, screen time, indoor playing time, outdoor playing time, and going out time) and the questionnaire data. Our findings revealed a relationship between physical activity and the neighborhood environment; however, the R2 values were low. Despite low prediction accuracy, the results of this study were similar to those of previous studies, indicating a positive correlation of physical activity with the absence of undulating roads, a crime-free environment, and safety around the home. Overall, these findings emphasize the need for providing children with opportunities for outdoor physical activity and play in the context of the COVID-19 recovery phase for adherence to the relevant guidelines.

## Introduction

A novel coronavirus infection, COVID-19, caused by a recently discovered coronavirus called SARS-CoV-2, was initially reported in Wuhan, China, in December 2019 and rapidly disseminated, exerting a profound impact on human health. Several nations implemented lockdown measures to mitigate infection transmission. In Japan, the government announced an emergency declaration requesting people to refrain from going outside and to avoid "crowded

**Funding:** The authors received no specific funding for this work.

**Competing interests:** The authors have declared that no competing interests exist.

places," "closed spaces," and "close contact with other people" (known as the 3Cs). Specific guidelines were issued and necessary actions were undertaken.

These measures have been documented to cause a decline in physical activity (PA) [1], escalation in obesity rates, and decrease in physical fitness levels [2, 3] in healthy adults. A systematic review conducted by Kharel et al. [4] ascertained that the measures implemented to prevent the spread of COVID-19 had detrimental consequences, including physical deterioration and increased engagement with screens among children and adolescents worldwide. Additional reports have also substantiated the decline in physical fitness and function among school-aged children [5–9].

In Japan, the results of the 2021 National Survey of Physical Fitness, Exercise Capacity, and Exercise Habits [10] showed that the physical fitness of school-aged children has declined compared to 2019. The main reason for this decline was stated to be a decrease in exercise engagement and an increase in screen time, as well as a sharp increase in the prevalence of obesity, which had been a concern for some time prior. The limitation of exercise by COVID-19 may have contributed significantly to the decline in physical fitness. A study involving elementary school children aged 6–12 years in urban Japan reported that only 60%, 21.5%, and 68.7% of the recommended daily moderate-to-vigorous PA (MVPA), sedentary activity, and sleep duration guidelines, respectively, were met [11]. PA for preschoolers aged 3–5 years has been reported to be important because of its benefits for social skill development, motor skill development, and cognitive development [12]. However, among Japanese preschoolers, adherence to the World Health Organization's (WHO) 24-Hour Movement Guidelines for the Early Years [13] experienced a significant decrease during the COVID-19 epidemic compared to the pre-epidemic period [14]. A report from Canada indicated that only 4.8% of school-aged children adhered to exercise guidelines at the peak of the COVID-19 epidemic [15]. During the COVID-19 epidemic, neighborhood outdoor spaces such as residential streets, local parks, and traffic areas surrounding homes appear to play a pivotal role in fostering children's engagement in PA [16]. The built environment of neighborhoods has been reported as a factor promoting PA even before the COVID-19 epidemic [17, 18]. Thus, inadequate neighborhood environments may impede children's participation in health-promoting physical activities.

Although the relationship between PA and neighborhood environments during COVID-19 has been investigated, most studies have primarily focused on school-aged children and adults, while limited research has been conducted on preschoolers, for whom physical activity is important [19–22].

Therefore, the purpose of this study was to determine the relationship between PA time and the neighborhood environment on weekend day for preschoolers during the COVID-19 epidemic and to determine the amount of physical activity among preschoolers during the COVID-19 epidemic.

Research focusing on the relationship between PAtime and the neighborhood environment can provide valuable insights into how to design and improve community spaces to promote active lifestyles among preschoolers. In addition, investigating PA hours for preschool children during the COVID-19 epidemic is essential for developing effective intervention strategies.

## Methods

### Participants

This study was conducted between October 11 and November 7, 2021. Participants were recruited after clearly informing the parents of the purpose of the study and included children attending kindergartens, nurseries, and childcare centers in Hashimoto City, Wakayama Prefecture. The study was requested by 1,241 parents of preschoolers aged 3–5 years attending

three kindergartens, five nurseries, and nine childcare centers, and cooperation was obtained from 357 parents. This study was conducted remotely to prevent infection.

## Ethical approval statement

This study was conducted with approval from the Ethics Committee of Kio University and Hashimoto City Department of Children. The main purpose of the study was explained in writing, and all participants provided consent to participate by completing the survey and sending their responses to a 24 h PA record (24hPAR).

## Instrumentation

PA was assessed using 24hPAR, a method of measuring PA by filling in the daily PA on paper. This method has been validated for middle-aged and older adults and for young adults [23, 24] and is also capable of capturing behavioral content.

Basic data and information on the neighborhood environment were collected using questionnaires. We asked parents to report one weekend day (Saturday or Sunday) PA and related basic data of their preschool children. The inclusion criteria were as follows: 1. Fill in each item on the 24PAR without omission. 2. Fill in each item on the questionnaires without omission. 3. Both the 24hPAR and questionnaire were answered.

## Procedures

The 24hPAR was performed based on previous studies [24, 25]. The respondents were asked to:

1. Record their preschooler's weekend day behavior in 15-minute increments

2. Record their activities freely.

3. Record the days of no trips or other events

4. Record on sunny weekend days

5. Record the date of recording

6. Fill the blanks in the 24hPAR

7. Record whether the activity was performed while sitting or standing

8. Record whether the activity was performed indoors or outdoors

We divided PA into nine separate categories (sedentary time (SET), sleeping hours (SH), screen time (SCT), indoor playing time (IP), outdoor playing time (OP), going out time (GO), extracurricular activity time (EAT), activity of daily living time (ADLT), and other). In our paper, we defined the categories as follows: SET [26] was defined as the time spent on sedentary activities (watching TV, reading books, etc.), excluding the SH and ADLT. This calculated the cumulative time of SET performed throughout the day.; SH was defined as sleeping time, including nap time; SCT [27] was defined as the time spent watching TV, playing games, or using tablets; IP was defined as the time spent on indoor activities (reading books and playing with blocks), excluding SCT; OP [28] was defined as the time spent on physical activities, such as playing in the park, tag, and sports(includes both play fun and competitive); GO was defined as physical activities, such as walking and shopping; and EAT was defined as a sedentary activity such as programming classes and cram school et al; finally, ADLT was defined as physical

activities, such as eating, bathing, and changing of clothes. Activities outside these eight categories were categorized as "other."

A questionnaire was used to collect basic data on the neighborhood environment. The questionnaire was administered in two patterns: via a Google form or on paper. The basic data questions were related to the preschoolers' current place of residence, grade of medical care, height, weight, age, sex, pre-existing medical conditions, history of hospitalization, medical history, exercise habits, vaccination status of the parents, and efforts to exercise. Questions regarding the neighborhood environment were developed based on studies conducted in Japan [22].

### Data analysis

All statistical analyses were performed using SPSS (IBM SPSS Statistics V25.0). After confirming the data normality, we used Spearman's rank correlation coefficient to analyze the relationship among five PA measures [SET, SCT, IP, OP, GO], and the questionnaire. The PA measures for SET, SCT, IP, OP, and GO were used as dependent variables. Variables found to be significantly related to each parameter were measured using Spearman's rank correlation coefficient, and items that were found to be related in previous studies [17] were used as independent variables. Stepwise multiple regression analysis was used to analyze these variables. Questionnaire items related to SET, SCT, IP, OP, and GO were examined. Analysis of variance was used to test the significance of multiple regression models. Additionally, the normality of residuals was confirmed. The Variance Inflation Factor (VIF) of all independent input variables was confirmed to be less than 10 to avoid multicollinearity problems. Statistical significance was set at $P < 0.05$.

## Results

357 parents were asked to complete the survey, out of which 187 completed it.

Table 1 shows the general characteristics of the study population. All the preschoolers were generally healthy and had no medical conditions that required consideration for play. As preschoolers were grouped at the facility according to age, we correspondingly divided our subjects into three groups: three-, four-, and five-year-olds.

### Physical activity time

Table 2 presents the PA times on holidays and their percentages. SET was calculated as a percentage by the number of minutes awake (1440 minutes minus average sleep hours).

Table 3 shows the results of the questionnaire, and Table 4 shows the correlation coefficients between the questionnaire results and the PA duration of the preschoolers on a weekend day. Significant differences were defined as $P < 0.05$.

The following three variables were significantly correlated with SET: (1) Gender, (2) "Screen time: has your current screen time changed compared to that in January 2020 (before the COVID-19 pandemic)," and (3) "do the roads around your home have many ups and downs." Six variables showed significant correlations with SCT: (1) "Screen time: has your current screen time changed compared to that in January 2020 (before the COVID-19 pandemic)," (2) "does your home have a garden," (3) time required to walk to the facilities, (4) "do you use the facilities with your children," (5) "do you feel unsafe around your home because of heavy traffic," and (6) "do the roads around your home have many ups and downs." The only variable revealing a significant correlation with IP was "are the facilities available on weekend days." The only variable showing a significant correlation with OP was "Screen time: How do you compare now with January 2020 (before the COVID-19 pandemic)?"Finally, the only

**Table 1. The general characteristics of study population (N = 187).**

|  | Overall, N, Average (%) |
|---|---|
| Age | 4.9 ± 10.3 |
| Boys | 107 (57.2) |
| Girls | 80 (42.8) |
| Kaup Index | 15.5 ± 1.5 |
| Skinny (BMI under 14) | 25 (13.4) |
| Normal (BMI 15–17) | 151 (80.7) |
| Obesity (BMI over 18) | 11 (5.9) |
| **Current Health** |  |
| Healthy | 172 (92.0) |
| Healthy, but attending hospital (No disruption in daily life, but is attending the hospital.) | 15 (8.0) |
| **Health Status for Exercise** |  |
| No medical conditions requiring consideration for exercise | 187 (100) |
| **Age group** |  |
| 3-year-olds | 64 (34.2) |
| 4-year-olds | 68 (36.4) |
| 5-year-olds | 55 (29.4) |
| **Facility** |  |
| Child care centers | 134 (71.7) |
| Nurseries | 13 (6.9) |
| Kindergartens | 40 (21.4) |

variable that showed a significant correlation with GO was the parents' (respondents') vaccination status against COVID-19.

The correlation coefficients between PA time and the questionnaire were determined. For SET, "Screen time: How do you compare now with January 2020 (before the COVID-19 pandemic)" showed r = 0.153, while "do the roads around your home have many ups and downs" showed r = 0.153. For SCT, "Screen time: How do you compare now with January 2020 (before the COVID-19 pandemic)" showed r = 0.237, indicating a high correlation coefficient in this study; further, "do the roads around your home have many ups and downs" also showed r = 0.234. For IP, r = 0.139 was obtained for "is the area around your home rich in nature" and

**Table 2. Physical activity time (N = 187).**

|  | Minutes ± SD |
|---|---|
| Sedentary time (SET) | 274.8 ± 107.1 (35.1%) |
| Sleeping hours(SH) | 656.4 ± 56.6 (45.6%) |
| Screen time (SCT) | 151.9 ± 84.5 (10.5%) |
| Indoor playing time (IOP) | 181.3 ± 87.9 (12.6%) |
| Outdoor playing time (OP) | 128.1 ± 93.2 (8.9%) |
| Going out time (GO) | 104.3 ± 87.0 (7.2%) |
| Extracurricular activity time (EAT) | 3.2 ± 17.0 (0.2%) |
| Activity of daily living time (ADLT) | 188.4 ± 46.5 (13.1%) |
| Other | 26.5 ± 32.5 (1.9%) |

**Table 3. Result of questionnaire (N = 187).**

| Items | Options | Range | N |
|---|---|---|---|
| ≪COVID-19≫ | | | |
| Parents' (respondents') vaccination status against COVID-19 | 1. I have completed my second dose | | 131 |
| | 2. I have completed my first dose | | 11 |
| | 3. Not yet | | 45 |
| Physical activity time: Has your current PA time changed compared to January 2020 (before the COVID-19 pandemic)? | 1. Very increased | | 10 |
| | 2. A bit increased | | 23 |
| | 3. Still the same | | 76 |
| | 4. A bit decreased | | 67 |
| | 5. Very decreased | | 11 |
| Sedentary time: Has your current SET changed compared to January 2020 (before the COVID-19 pandemic)? | 1.Very decreased | | 0 |
| | 2. A bit decreased | | 9 |
| | 3.Still the same | | 83 |
| | 4. A bit increased | | 82 |
| | 5. Very increased | | 13 |
| Screen time: Has your current SCT changed compared to January 2020 (before the COVID-19 pandemic)? | 1. Very decreased | | 0 |
| | 2. A bit increased | | 3 |
| | 3.Still the same | | 61 |
| | 4. A bit increased | | 79 |
| | 5. Very increased | | 44 |
| Did you have any ideas on how to get your child to exercise? | 1.Yes | | 123 |
| | 2.No | | 61 |
| ≪About your home≫ | | | |
| What is your type of residence? | 1. Detached house | | 160 |
| | 2. Apartment building | | 27 |
| How many rooms? | 1. Numerous | (6–10) | 23 |
| | 2. Moderate | (4–5) | 133 |
| | 3. Few | (2–3) | 25 |
| (only lived in a detached house) Does your home have a garden? | 1. Yes | | 151 |
| | 2. No | | 9 |
| (Only for those living in an apartment building) How many floors? | 1. Low | (2–3) | 15 |
| | 2. Moderate | (4–12) | 8 |
| | 3. High | (13–32) | 4 |
| (Only live in an apartment building) What floor do you live on? | 1. Low | (1–2) | 14 |
| | 2. Moderate | (3–5) | 7 |
| | 3. High | (6–15) | 6 |
| Do you own a car? | 1. Yes | | 186 |
| | 2. No | | 1 |
| ≪Regarding the neighborhood environment≫ | | | |
| Is there any park in the neighborhood? | 1. Yes | | 175 |
| | 2. No | | 12 |
| Are there any sports facilities in the neighborhood? | 1. Yes | | 108 |
| | 2. No | | 79 |

(*Continued*)

**Table 3.** (Continued)

| Items | Options | Range | N |
|---|---|---|---|
| (Only answered by those who have facilities) Access to the facility | 1. Good | | 94 |
| | 2. Not good | | 14 |
| (Only answered by those who have facilities) Time required to walk to the facilities | 1. Near | (0.5–2.5) | 7 |
| | 2. Moderate | (3–10) | 30 |
| | 3. Far | (13–240) | 71 |
| (Only answered by those who have facilities) Time it takes to drive to the facility | 1. Near | (0.5–3) | 30 |
| | 2. Moderate | (4–7) | 45 |
| | 3. Far | (10–30) | 33 |
| (Only answered by those who have facilities) Are the facilities available for use during holidays? | 1. Yes | | 102 |
| | 2. No | | 6 |
| (Only answered that have facilities) Is the facility available at night? | 1. Yes | | 44 |
| | 2. No | | 64 |
| (Only answered by those who have facilities) Parents only use the facility. | 1. No | | 104 |
| | 2. Yes | | 4 |
| (Only answered by those who have facilities) Do you use the facilities with your children? | 1. Yes | | 79 |
| | 2. No | | 29 |
| (Only answered by those who have facilities) Satisfaction with the facility | 1. Satisfaction | | 85 |
| | 2. Unsatisfied | | 23 |
| Are there any sidewalks around your home? | 1. Yes | | 140 |
| | 2. No | | 47 |
| Do you feel unsafe around your home due to heavy traffic? | 1. Yes | | 86 |
| | 2. No | | 101 |
| There are few streetlights around your home. | 1. Yes | | 70 |
| | 2. No | | 117 |
| Are the roads around your home with many ups and downs? | 1. Yes | | 100 |
| | 2. No | | 87 |
| Is the area around your home crime-free and safe? | 1. Yes | | 162 |
| | 2. No | | 25 |
| Is there abundant nature around your home? | 1. Yes | | 182 |
| | 2. No | | 5 |
| Is there a view around your home that your parents like? | 1. Yes | | 153 |
| | 2. No | | 34 |
| Is public transportation easily accessible from your home? | 1. Yes | | 90 |
| | 2. No | | 97 |
| Do your children refrain from exercise, sports, or going out when the weather is bad? | 1. Yes | | 11 |
| | 2. No | | 176 |
| Do your children refrain from exercise, sports, or going out depending on the season? | 1. Yes | | 65 |
| | 2. No | | 122 |

r = 0.104 was obtained for "is there a view that the parents like being around the home." For OP, the correlation coefficients were r = 0.116 for "age," r = 0.106 for "only parents use the facility" and r = 0.118 for "do you use the facilities with your children." GO showed low correlation coefficients for all the variables.

**Table 4. Correlation analysis: Physical activity time and questionnaire (N = 187).**

|  |  | SET | SCT | IP | OP | GO |
|---|---|---|---|---|---|---|
| Date of entry | r values | 0.135 | 0.067 | −0.106 | 0.034 | −0.027 |
|  | P values | 0.065 | 0.361 | 0.149 | 0.643 | 0.712 |
| Facilities | r values | −0.010 | −0.018 | −0.018 | −0.027 | 0.020 |
|  | P values | 0.896 | 0.803 | 0.807 | 0.718 | 0.786 |
| Gender | r values | −0.173 | −0.136 | −0.031 | −0.013 | 0.051 |
|  | P values | 0.018 | 0.063 | 0.674 | 0.861 | 0.487 |
| Age | r values | 0.023 | 0.122 | −0.122 | 0.116 | 0.022 |
|  | P values | 0.753 | 0.097 | 0.095 | 0.113 | 0.765 |
| Parents'(respondents')vaccination status against COVID-19 | r values | −0.038 | 0.016 | 0.076 | 0.028 | −0.146 |
|  | P values | 0.608 | 0.829 | 0.302 | 0.699 | 0.046 |
| Physical activity time: How do you compare now with January 2020 (before the COVID-19 pandemic)? | r values | 0.025 | 0.082 | 0.005 | −0.110 | −0.062 |
|  | P values | 0.730 | 0.264 | 0.945 | 0.135 | 0.398 |
| Sedentary time: How do you compare now with January 2020 (before the COVID-19 pandemic)? | r values | 0.091 | 0.080 | −0.020 | −0.111 | −0.040 |
|  | P values | 0.217 | 0.274 | 0.790 | 0.131 | 0.583 |
| Screen time: has your current screen time changed compared to that in January 2020 (before the COVID-19 pandemic), | r values | 0.153 | 0.237 | −0.122 | −0.176 | 0.026 |
|  | P values | 0.036 | 0.001 | 0.096 | 0.016 | 0.723 |
| Did you have any ideas on how to get your child to exercise? | r values | 0.092 | 0.056 | 0.021 | −0.133 | −0.026 |
|  | P values | 0.209 | 0.447 | 0.771 | 0.069 | 0.722 |
| Which is your type of residence? | r values | 0.046 | −0.050 | 0.076 | −0.063 | −0.104 |
|  | P values | 0.536 | 0.499 | 0.299 | 0.395 | 0.157 |
| How many rooms? | r values | 0.047 | −0.032 | 0.026 | −0.080 | 0.048 |
|  | P values | 0.523 | 0.662 | 0.723 | 0.274 | 0.515 |
| Do you own a car? | r values | 0.039 | 0.033 | 0.005 | −0.033 | −0.001 |
|  | P values | 0.592 | 0.657 | 0.948 | 0.650 | 0.993 |
| (Only lived in a detached house) Does your home have a garden? | r values | 0.044 | 0.156 | −0.058 | 0.015 | 0.000 |
|  | P values | 0.585 | 0.048 | 0.464 | 0.853 | 0.997 |
| (Only live in an apartment building) How many floors? | r values | 0.076 | 0.214 | −0.160 | −0.001 | −0.052 |
|  | P values | 0.699 | 0.273 | 0.416 | 0.997 | 0.793 |
| (Only live in an apartment building)What floor do you live on? | r values | 0.135 | 0.098 | 0.094 | 0.034 | −0.145 |
|  | P values | 0.494 | 0.618 | 0.633 | 0.866 | 0.462 |
| Is there a park in the neighborhood? | r values | 0.052 | 0.004 | 0.041 | −0.082 | −0.005 |
|  | P values | 0.476 | 0.955 | 0.579 | 0.263 | 0.941 |
| Are there any sports facilities in the neighborhood? | r values | 0.115 | 0.038 | 0.058 | −0.074 | −0.037 |
|  | P values | 0.119 | 0.609 | 0.429 | 0.314 | 0.619 |
| (Only answered that have facilities) Access to the facility | r values | −0.094 | −0.070 | 0.002 | −0.139 | 0.024 |
|  | P values | 0.336 | 0.473 | 0.985 | 0.154 | 0.807 |
| (Only answered that have facilities) Time required to walk to the facilities | r values | −0.177 | −0.270 | −0.035 | −0.009 | 0.090 |
|  | P values | 0.068 | 0.005 | 0.722 | 0.924 | 0.358 |
| (Only answered that have facilities) Time it takes to drive to the facility | r values | −0.110 | −0.117 | −0.080 | −0.065 | 0.009 |
|  | P values | 0.258 | 0.231 | 0.414 | 0.509 | 0.924 |
| (Only answered that have facilities) are the facilities available on weekend days | r values | 0.133 | 0.104 | −0.222 | 0.096 | 0.055 |
|  | P values | 0.176 | 0.290 | 0.023 | 0.329 | 0.578 |
| (Only answered that have facilities)Is the facility available at night? | r values | 0.123 | 0.041 | 0.034 | −0.185 | −0.089 |
|  | P values | 0.210 | 0.678 | 0.734 | 0.059 | 0.366 |

*(Continued)*

**Table 4.** (Continued)

| | | SET | SCT | IP | OP | GO |
|---|---|---|---|---|---|---|
| (Only answered that have facilities) Parents only use the facility. | r values | −0.050 | 0.007 | −0.084 | 0.106 | −0.030 |
| | P values | 0.609 | 0.940 | 0.393 | 0.278 | 0.761 |
| (Only answered that have facilities) Do you use the facilities with your children? | r values | −0.171 | −0.192 | −0.084 | 0.118 | 0.035 |
| | P values | 0.079 | 0.048 | 0.393 | 0.227 | 0.723 |
| (Only answered that have facilities) Satisfaction with the facility | r values | −0.117 | −0.187 | 0.067 | −0.099 | 0.061 |
| | P values | 0.243 | 0.059 | 0.501 | 0.322 | 0.542 |
| Are there any sidewalks around your home? | r values | −0.060 | 0.066 | −0.026 | −0.065 | 0.001 |
| | P values | 0.416 | 0.371 | 0.720 | 0.378 | 0.985 |
| Do you feel unsafe around your home because of heavy traffic | r values | 0.123 | 0.148 | 0.088 | −0.072 | −0.076 |
| | P values | 0.092 | 0.043 | 0.232 | 0.329 | 0.302 |
| There are few street lights around your home. | r values | 0.103 | 0.076 | 0.009 | −0.051 | 0.016 |
| | P values | 0.159 | 0.298 | 0.898 | 0.488 | 0.832 |
| Do the roads around your home have many ups and downs. | r values | 0.153 | 0.234 | 0.002 | −0.039 | −0.088 |
| | P values | 0.037 | 0.001 | 0.975 | 0.592 | 0.230 |
| Is the area around your home crime-free and safe? | r values | 0.012 | 0.000 | −0.111 | 0.063 | 0.033 |
| | P values | 0.866 | 0.995 | 0.130 | 0.392 | 0.652 |
| Is the area around your home rich in nature | r values | 0.014 | −0.019 | 0.139 | −0.087 | 0.041 |
| | P values | 0.851 | 0.795 | 0.057 | 0.235 | 0.574 |
| Is there a view that the parents like being around the home? | r values | 0.080 | −0.078 | 0.104 | −0.086 | −0.069 |
| | P values | 0.277 | 0.289 | 0.156 | 0.240 | 0.350 |
| Is public transportation easily accessible from your home? | r values | −0.014 | −0.013 | −0.079 | −0.018 | 0.080 |
| | P values | 0.851 | 0.864 | 0.284 | 0.812 | 0.278 |
| Do your children refrain from exercise, sports, or going out when the weather is bad? | r values | −0.062 | 0.091 | −0.013 | 0.020 | −0.073 |
| | P values | 0.396 | 0.217 | 0.863 | 0.787 | 0.319 |
| Do your children refrain from exercise, sports, or going out depending on the season? | r values | −0.042 | 0.005 | 0.009 | 0.004 | −0.033 |
| | P values | 0.570 | 0.950 | 0.908 | 0.957 | 0.649 |

SET = sedentary time

SCT = screen time

IP = Indoor playing time

OP = outdoor playing time

OG = going out time

r values = correlation coefficient

P values = probability-value

## Multiple regression analysis

We used variables that were significantly related to each PA as measured using Spearman's rank correlation coefficient, and those that were found to be related in previous studies [17] as independent variables. We performed a stepwise multiple regression analysis to analyze these variables.

The factors influencing each category were also determined. SET was influenced by sex and "do the roads around your home have many ups and downs." SCT was influenced by "do the roads around your home have many ups and downs" and "Screen time: How do you compare now with January 2020 (before the COVID-19 pandemic)." IP was influenced by "is the area around your home crime-free and safe." OP was influenced by "Screen time: How do you

**Table 5. Multiple regression analysis (N = 187).**

| Item | P-values |
|---|---|
| Sedentary time (R2 = 0.116) | |
| Are the roads around your home with many ups and downs? | 0.017 |
| Gender | 0.039 |
| Screen time (R2 = 0.107) | |
| screen time: compared to January 2020 (before the COVID-19) | 0.000 |
| Are the roads around your home with many ups and downs? | 0.007 |
| Indoor playing time (R2 = 0.037) | |
| Is the area around your home crime-free and safe? | 0.024 |
| Outdoor playing time (R2 = 0.033) | |
| screen time: compared to January 2020 (before the COVID-19 outbreak) | 0.012 |
| Going out time (R2 = 0.032) | |
| Parents' (respondents') vaccination status against COVID-19 | 0.048 |
| What floor do you live on? | 0.031 |

R2 = The coefficient of determination

compare now with January 2020 (before the COVID-19 pandemic)." GO was influenced by the parents' (respondents') vaccination status against COVID-19 and "on what floor do you live."

Table 5 shows the results of multiple regression analysis. All multiple regression models were significant ($P < 0.05$), and VIF was less than 10, indicating no multicollinearity problems. The coefficient of determination ($R^2$) was 0.116 for SET, 0.107 for SCT, 0.037 for IP, 0.033 for OP, and 0.032 for GO.

Although a relationship was found, the R2 values were low for all dependent variables, resulting in low R2 values.

## Discussion

### Physical activity time

In this study on the PA of preschoolers, five-time periods were extracted from the 24PAR and analyzed. The WHO guidelines recommend [13] that (a) 3–4 years old children should spend at least 180 minutes in various types of physical activities at any intensity; (b) sedentary screen time should be no more than 1 h (the less time spent, the better); and (c) 3–4 years old children should have 10–13 hours of good quality sleep. By limiting the number of children included in this study to those aged 3–4 years, we found that: (a) 35 of 132 (26.5%), (b) 14 of 132 (10.6%), and (c) 125 of 132 (94.7%) children met the respective WHO guidelines. SCT was 2.5 times higher than that in the WHO guidelines [13].

Children typically obtain their daily PA through organized sports, active play, and by spending time in playgrounds and parks. Conversely, most of their sedentary time accumulated at home [29]. Children spent less time outdoors during the COVID-19 epidemic [30]. After the COVID-19 outbreak, parents' self-implemented restrictions on their children, owing to the risk of infection, resulted in less GO and increased SCT at home [31]. These limitations may have led to lower PA and ultimately increased the SCT in this study. SCT compliance is reported to be more strongly associated with family factors such as parents' modeling of SCT, parents' TV viewing, and house rules, as opposed to personal or social factors [32]. Similarly, parents themselves may experience decreased PA and increased SCT. The importance of the

interaction between exercise and non-exercise activities should not be considered in isolation [33]. Preschoolers in particular, need to be more inclusive with respect to PA, as they may be influenced by their parents. Adherence to the amount of PA and WHO guidelines during the COVID-19 epidemic in preschoolers has been reported [31, 34], but as far as we have been able to determine, no reports have provided detailed behavioral details. Preschoolers may engage in many types of PA. In studies with preschoolers, recording not only the amount of PA but also the content of behavior may assist in planning interventions for PA.

## Multiple regression analysis

Multiple regression analysis showed that the $R^2$ values were lower for all PA durations used as dependent variables. Smith, et al. [35] in their review of the relationship between PA and neighborhood environment for children aged 5–13 years, reported that a well-connected location, high population density, proximity to various destinations, short commuting distances, and a safe transportation environment are important factors that support active travel behavior in children. This study also found that the absence of crime and safety around the home, an abundance of nature, and lack of ups and downs on the roads around the home were related to a positive relationship between PA and the neighborhood environment among preschoolers in Japan [17]. Regarding the relationship between PA and neighborhood environment in children aged 5–11 years during the COVID-19, a lower residential density and areas farther from arterial roads were found to be associated with increased outdoor activity, whereas having a park within 1 km of the residence was negatively associated with increased outdoor activity [16]. Although R2 values were low, the relationship between PA and neighborhood environment in this study was correlated with the lack of severely undulating roads and crime-free and safe home locations. The results of this study are similar to those of previous studies. However, the lower R2 values may have resulted from the special circumstances of the COVID-19 epidemic. The low R2 value suggests that the relationship between PA and neighborhood environment is low among Japanese preschoolers, and may be similarly low in other areas of Japan. few studies have examined the relationship between PA and neighborhood environment among COVID-19 preschoolers during the COVID-19 epidemic, and this study may inform research on preschoolers after the COVID-19 epidemic, this study may be useful for future studies of preschoolers after the COVID-19 epidemic.

## Suggestions for increasing PA

In May 2023, COVID-19 was downgraded to a "common infectious disease" in Japan. In the context of the COVID-19 recovery phase, providing children with opportunities for outdoor PA and play is increasingly important. Parental involvement may further promote PA. Children who play outdoors tend to be more active, sit less often, and sleep better; therefore, opportunities to play outdoors increase the likelihood of meeting the PA guidelines [36].

## Limitations

This study has several limitations. First, this evaluation was limited to only one day during a weekend day to minimize the relevant burden on parents and our data were limited because the study was conducted during a weekend day to allow for a full day of observing the preschoolers. Therefore, it is difficult to generalize our findings based solely on these data. Future research should also consider evaluation methods that do not require parental supervision.

Second, the use of accelerometer-based assessment, which is more valid and reliable than the questionnaire method in comparison to the gold standard double-labeled water method in

measuring PA, should also be considered. However, the 24PAR used in this study is useful for accurately assessing physical activity at a low cost.

## Conclusions

The R2 values were low, although the relationship between gender, changes in screen time compared to pre-COVID-19, road ups and downs, and traffic around the home was evident. Although the results of this study are similar to those of previous studies, the low predictive accuracy may be attributed to the unique circumstances of the COVID-19 epidemic. This study focused on preschoolers' PA and neighborhood environment, and the results may contribute to future research on the long-term effects of preschoolers after the COVID-19 epidemic.

## Acknowledgments

We would like to thank the parents of young children attending kindergartens, nurseries, and childcare centers in Hashimoto City, and the staff of the Department of Children.

I would like to thank Namba.H for helpful discussions.

The authors would like to thank editage (https://www.editage.jp/) for the English language review.

## Author Contributions

**Conceptualization:** NAOKI YAMAGUCHI, TAKAHIKO FUKUMOTO.

**Data curation:** NAOKI YAMAGUCHI, TAKAHIKO FUKUMOTO.

**Formal analysis:** NAOKI YAMAGUCHI, TAKAHIKO FUKUMOTO.

**Funding acquisition:** NAOKI YAMAGUCHI, TAKAHIKO FUKUMOTO.

**Investigation:** NAOKI YAMAGUCHI, TAKAHIKO FUKUMOTO, HIDETAKA IMAGITA.

**Methodology:** NAOKI YAMAGUCHI, TAKAHIKO FUKUMOTO, HIDETAKA IMAGITA.

**Project administration:** NAOKI YAMAGUCHI, HIDETAKA IMAGITA.

**Resources:** NAOKI YAMAGUCHI.

**Software:** NAOKI YAMAGUCHI.

**Supervision:** NAOKI YAMAGUCHI, TAKAHIKO FUKUMOTO, HIDETAKA IMAGITA.

**Validation:** NAOKI YAMAGUCHI, TAKAHIKO FUKUMOTO, HIDETAKA IMAGITA.

**Visualization:** NAOKI YAMAGUCHI.

**Writing – original draft:** NAOKI YAMAGUCHI.

**Writing – review & editing:** NAOKI YAMAGUCHI, TAKAHIKO FUKUMOTO, HIDETAKA IMAGITA.

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
