## [Decision Letter · Decision Letter 0]

5 Mar 2024

PONE-D-23-22742Relationship between physical activity and neighborhood environment in preschool children during COVID-19. - A cross-sectional study using 24-hour activity records-PLOS ONE

Dear Dr. Yamaguchi,

Thank you for submitting your manuscript to PLOS ONE. After careful consideration, we feel that it has merit but does not fully meet PLOS ONE’s publication criteria as it currently stands. Therefore, we invite you to submit a revised version of the manuscript that addresses the points raised during the review process.

We look forward to receiving your revised manuscript.

Kind regards,

Patrick WC Lau

Academic Editor

PLOS ONE

Journal Requirements:

4.Thank you for stating the following financial disclosure:

"No"

Additional Editor Comments:

The comments and suggestions from the three reviewers are precise and meaningful for the authors to revise the manuscript. English editing is highly encouraged to improve the paper clarity for resubmission.

Reviewers' comments:

Reviewer's Responses to Questions

**Comments to the Author**

1. Is the manuscript technically sound, and do the data support the conclusions?

Reviewer #1: Partly

Reviewer #2: Yes

Reviewer #3: Partly

2. Has the statistical analysis been performed appropriately and rigorously? 

Reviewer #1: Yes

Reviewer #2: Yes

Reviewer #3: I Don't Know

3. Have the authors made all data underlying the findings in their manuscript fully available?

Reviewer #1: No

Reviewer #2: Yes

Reviewer #3: Yes

4. Is the manuscript presented in an intelligible fashion and written in standard English?

Reviewer #1: Yes

Reviewer #2: Yes

Reviewer #3: No

5. Review Comments to the Author

Reviewer #1: The manuscript titled “Relationship between physical activity and neighborhood environment in preschool children during COVID-19” describes possible relationships between the physical environment where these children live (in Hashimoto City, Wakayama Prefecture, Japan). The premise of the study was that the measures taken to prevent the spread of COVID-19 would impact physical activity patterns for these preschoolers (ages 3-5). The authors used a multiple regression model to analyze the environmental factors that could predict decline in physical activity. The outcomes reflect their conclusion, namely that the environment plays a critical role in physical activity time. However, the study was premised on the changes in physical activity caused by measures aimed at preventing the spread of the virus; to that end, the study failed to demonstrate conclusively that restrictions caused by COVID-19 were responsible for decline in physical activity by these children. This fundamental point could perhaps be made if this reviewer had access to the actual data from the national survey (ref. 10, https://www.mext.go.jp/sports/b_menu/toukei/kodomo/zencyo/1411922_00003.html), alas the language barrier prevented me from being able to identify such patterns.

Given the concerns identified above, the study does not offer any novel elements not previously identified in the available literature, cited in the manuscript and elsewhere. I would understand why this age group need to be studied, if the authors had provided some evidence as to the importance of physical activity in this age cohort to establish healthy lifestyle habits. Further, there are a few critical methodological concerns that prevent conclusions specific to the population under study (preschoolers ages 3-5). I shall mention a few of these in the sections that follow.

1. Lines 62-74 include data indicating that physical activity declined during COVID; yet, in lines 74-76 the authors contend than “…children have augmented their PA duration by engaging in diverse combinations of physical activities.” Such a conclusion was confusing and totally unsubstantiated.

2. The notion that the neighborhood environment may be conducive or preventive towards regular physical activity is not new, thus the mention in lines 78-83 is unnecessary. However, if the intent was to show that specific environments encouraged PA despite COVID, this was never expressed in the purpose statement at the end of the introduction.

3. The authors use the 24hPAR method to record physical activity data for the preschoolers, yet in the Limitstions section of the paper they indicate that “…a better evaluation method should be considered in future studies.”, in effect negating the outcomes of this paper.

4. Ultimately, the lack of statistical significance and low predictive validity negates any possible relationship between physical activity and determining environmental factors. Thus, it is difficult to ascertain what the main message of this paper would be, other that the issue of whether children engage in regular physical activity or not is multifaceted.

I appreciate the effort the authors made to improve the manuscript based on the comments of the previous reviewer. However, I consider some elements of the paper to require additional attention. I will mention a few of these issues below.

1. Line 71: ‘…compliance with guidelines witnessed a substantial decline…” awkward phrasing: compliance cannot witness anything…

2. The term “holiday” was retained but should be characterized “a weekend day”.

3. In the Participants section of the Method, there is no clarity as to how many days data were recorded (One weekend day? Both weekend days? All four weekends?). That is mentioned in the Limitations section of the paper.

4. In the Instrumentation the authors indicate that recording was done at 1-minute increments (line 108), yet in the next section they mention 15-minute increments (line 119).

5. The division of physical activity (lines 160-171) should be part of the introduction, as this would be the place where the authors would identify which measures of physical activity would be relevant to their study.

6. The total number of complete surveys (n=187) needs to be reported in the Results section.

7. Lines 152-154 and 154-155 are repetitive.

8. Data need not be reported both in the text and in the Tables.

9. Lines 180-182: why are these included in the subheading about physical activity time?

10. Line 204: I believe the table lists an r=0.234 not r=0.237 as written.

11. Line 299: restructure the sentence “… but with low predictive accuracy was low.”

Reviewer #2: Thank you for your scientific contribution. I have some concerns which need to be adressed:

1) Did you preregister the study? For example on OSF? Furthermore, no specific guidelines were followed such as STROBE for example.

2) Line 143-144:...and items that were found to be related in previous studies were used as independent variables. Which studies do you mean?

3) Were independence, variance homogeneity and normal distribution of residues tested for multiple regression?

4) Line 153-155: Too many word repetitions

5) Line 160-171: Should be in the methods section

6) Line 171: "Table 2 presents the PA times on holidays and their percentages." I cannot find any percentages

7) Line 174-179: Should be integrated in discussion section

8) Line 174: [19] appears to be cited wrongly

9) Line 182: Did you mean 0.05 istead of 0.5? You did already state that in the methods section.

10) Line 298-299: Reword this sentence

11) Line 301: Use "pandemic" instead of "epidemic"

Reviewer #3: I thank the authors for sharing their work “Relationship between physical activity and neighborhood environment in preschool children during COVID-19.” I appreciate the opportunity to be able to review their work and would like to provide suggestions for major revisions. I have provided detailed comments below. I have noted several grammatical, punctuation, and formatting concerns in the manuscript and tables – correcting these would enhance the overall quality of the manuscript. Throughout the manuscript, some sentences remain fragmented so it is challenging to infer logical connections.

I hope that my feedback is helpful and would be happy to provide additional clarification as needed.

Abstract

- Line 26 – please consider elaborating on what is meant by “optimal health”

- Line 29 – does the term “holidays” refer to weekends (Saturday and/or Sunday) in the context of this paper? If yes, I would suggest potentially changing “holidays” to “weekend” throughout your manuscript.

- Line 32 – please consider starting your sentence with “Physical activity” instead of the acronym “PA.”

Introduction

- Line 49 – the authors refer to the coronavirus as “exerting a profound impact on human health.” Can you please elaborate on this to more specifically state what aspects of human health, wellbeing, mental health, etc. were impacted by coronavirus? I think this sentence could be strengthened with the additions of a citation.

- Lines 55-56 – can you please clarify if these trends were observed in adults or in children and adolescents?

- Lines 62-66 – I appreciate the authors referencing a national physical fitness and exercise survey, however, these sentences could be improved for greater clarity. What deteriorated and what population (e.g., adults, children, adolescents) is this referring to?

- Line 67 – I would suggest including a citation here.

- Line 71 – could you please clarify if the guidelines are referring to physical activity guideline or exercise guidelines?

- Lines 78-79 – did you operationally define “neighborhood environment” in the context of your study? Could you please revise this sentence to clarify what is meant by “the extent and nature of both PA and sedentary behavior”?

- Lines 87-88 – you provide background on the pandemic and its impact on physical activity, sedentary time, etc., however, there is no mention of the COVID-19 pandemic in your aim. If this is pivotal to your work and since it is mentioned in the article title, I would suggest revising your aim to include mention of the pandemic.

- I would suggest included a greater emphasis on the importance of including preschool aged children in research studies. You mention that this group is considered a vulnerable population earlier in the introduction – providing another sentence or two on the importance/significance of this work among preschoolers may help strengthen the introduction.

Methods

- Were parents formally trained on how to use the questionnaire or how to observe their children? If so, it would be helpful to describe and would enhance the quality of the methods.

- Is there any additional information available on the family of the subjects? For example, was it a two-parent household? Did the subject have younger or older siblings, or were they only children?

- Lines 107-108 – can you please clarify if a standardized questionnaire was used for the 24h PAR? If so, please include a citation for the questionnaire.

- Line 113 – can you please elaborate on what is meant by “deficiencies in the 24hPAR”?

Data Analysis

- Line 137 – please consider adding additional statistical software information.

- Lines 139-141 – I appreciate the authors’ efforts to include definitions for these variables. I would suggest including them in the methods section. Please also include a citation if this is how these variables are operationally designed in existing literature.

Results

- Lines 152-155 – these lines appear to be redundant; I would suggest revising for conciseness.

- Lines 160-161 – the authors do a nice job of defining PA variables earlier in the paper, however, it appears that sleeping hours, activity of daily living, and extracurricular activity time were not defined. I would consider defining these in the methods section with your other variables for greater clarity.

- Lines 163-170 – please consider including these definitions in the methods section as well as citations.

- Line 168 – please confirm if EAT should be defined as sedentary activity or elaborate to better distinguish it from ST.

- Lines 172-173 – please consider revising if you are intending to report mean±SD.

- Lines 174 and 176 – does “subjects” refer to children? If so, I would update to be as specific as possible.

- Line 182 – there appears to be a typo in this line for statistical significance and this is already stated in the statistical analysis section.

- Line 213 – there is a typo here.

- Line 214 – you mention previous studies – please include a citation.

Discussion

Overall, the discussion should be revised to be better aligned with the aim of this work. The discussion could be greatly strengthened by emphasizing the main findings of the relationship between physical activity and the environment among preschoolers. The discussion could be greatly improved by clearly and concisely comparing the current study with previous literature and emphasizing what is new or novel about this work.

Please consider revising lines 288-289 – this sentence is confusing.

Please consider revising the conclusion to include more specific results (Lines 298-299) with an emphasis on what this work may contribute to the existing literature.

Table 1 – Please consider adding sample size to the title. I would also suggest relabeling “Man” and “Women” as “Boys” and “Girls” or “Males” and “Females”, respectively. Can you please also confirm if “Skinny, Normal, Obesity” is referring to BMI? If so, please include units. I would also suggest clarifying what is meant by “Healthy, but attending hospital” by including reference to this label in a legend. Lastly, there appears to be a typo in “4-year old” as this should be plural.

For Tables 2, 3, 4 – please consider double checking for formatting/consistent formatting, spacing, etc. Presently, Table 4 in particular is challenging to read/interpret.

6. PLOS authors have the option to publish the peer review history of their article (what does this mean?). If published, this will include your full peer review and any attached files.

Reviewer #1: **Yes: **Stasinos Stavrianeas, Ph.D.

Reviewer #2: No

Reviewer #3: **Yes: **Nicole Bajdek

---

## [Author Response · Author response to Decision Letter 0]

1 May 2024

Review Comments to the Author

Reviewer #1: The manuscript titled “Relationship between physical activity and neighborhood environment in preschool children during COVID-19” describes possible relationships between the physical environment where these children live (in Hashimoto City, Wakayama Prefecture, Japan). The premise of the study was that the measures taken to prevent the spread of COVID-19 would impact physical activity patterns for these preschoolers (ages 3-5). The authors used a multiple regression model to analyze the environmental factors that could predict decline in physical activity. The outcomes reflect their conclusion, namely that the environment plays a critical role in physical activity time. However, the study was premised on the changes in physical activity caused by measures aimed at preventing the spread of the virus; to that end, the study failed to demonstrate conclusively that restrictions caused by COVID-19 were responsible for decline in physical activity by these children. This fundamental point could perhaps be made if this reviewer had access to the actual data from the national survey (ref. 10, https://www.mext.go.jp/sports/b_menu/toukei/kodomo/zencyo/1411922_00003.html), alas the language barrier prevented me from being able to identify such patterns.

RESPONSE： Thank you for dedicating your time and effort to reviewing my paper. Your feedback was incredibly helpful in improving the manuscript. I appreciate your thoughtful and constructive comments, which strengthened the research.

For ref.10, a PDF is attached to the other files.

Given the concerns identified above, the study does not offer any novel elements not previously identified in the available literature, cited in the manuscript and elsewhere. I would understand why this age group needs to be studied if the authors had provided some evidence as to the importance of physical activity in this age cohort to establish healthy lifestyle habits. Further, there are a few critical methodological concerns that prevent conclusions specific to the population under study (preschoolers ages 3-5). 

RESPONSE： Thank you for this suggestion. We have incorporated your comments and Inserted sentences on lines 71-73.

I shall mention a few of these in the sections that follow.

1. Lines 62-74 include data indicating that physical activity declined during COVID; yet, in lines 74-76 the authors contend than “…children have augmented their PA duration by engaging in diverse combinations of physical activities.” Such a conclusion was confusing and totally unsubstantiated.

RESPONSE： Thank you for this suggestion. We have incorporated your comments and deleted lines 74-78.

2. The notion that the neighborhood environment may be conducive or preventive towards regular physical activity is not new, thus the mention in lines 78-83 is unnecessary. However, if the intent was to show that specific environments encouraged PA despite COVID, this was never expressed in the purpose statement at the end of the introduction.

RESPONSE： Thank you for this suggestion. We agree with this comment and have added a new sentence for the purpose of the introduction(lines #92-#96).

3. The authors use the 24hPAR method to record physical activity data for the preschoolers, yet in the Limitstions section of the paper they indicate that “…a better evaluation method should be considered in future studies.”, in effect negating the outcomes of this paper.

RESPONSE： Thank you for your update. We have removed the text that we believe led to misinterpretation and added a new sentence for the Limitations section(lines #298-#299).

4. Ultimately, the lack of statistical significance and low predictive validity negates any possible relationship between physical activity and determining environmental factors. Thus, it is difficult to ascertain what the main message of this paper would be, other that the issue of whether children engage in regular physical activity or not is multifaceted.

RESPONSE：Thank you for this suggestion. The novelty and primary objective of this study lie in the absence of a discernible relationship between neighborhood environment and physical activity during COVID-19, contrary to previous findings that have reported such a relationship.

I appreciate the effort the authors made to improve the manuscript based on the comments of the previous reviewer. However, I consider some elements of the paper to require additional attention. I will mention a few of these issues below.

1. Line 71: ‘…compliance with guidelines witnessed a substantial decline…” awkward phrasing: compliance cannot witness anything…

RESPONSE： Thank you for this suggestion. Deleted pertinent text and filled in new text on lines 74-75.

2. The term “holiday” was retained but should be characterized “a weekend day”.

RESPONSE： Thank you for this suggestion. in the body of the manuscript to holiday in the text has been changed to "weekend day".

3. In the Participants section of the Method, there is no clarity as to how many days data were recorded (One weekend day? Both weekend days? All four weekends?). That is mentioned in the Limitations section of the paper.

RESPONSE： Thank you for this suggestion. We incorporated this comment and filled in "one weekend day" in the Instrumentation section of the method (line #120).

4. In the Instrumentation the authors indicate that recording was done at 1-minute increments (line 108), yet in the next section they mention 15-minute increments (line 119).

RESPONSE： Thank you for this suggestion. We have incorporated this input and the corresponding sentence in line 108 was deleted because line 119 describes recording in 15-minute increments.

5. The division of physical activity (lines 160-171) should be part of the introduction, as this would be the place where the authors would identify which measures of physical activity would be relevant to their study.

RESPONSE： Thank you for this suggestion. We have incorporated and filled in lines 160-171 has been inserted into the procedures section of the method(lines #137-#150).

6. The total number of complete surveys (n=187) needs to be reported in the Results section.

RESPONSE： Thank you for this suggestion. We have incorporated the number of participants who completed the research in the Results section (lines #174).

7. Lines 152-154 and 154-155 are repetitive.

RESPONSE： Thank you for this suggestion. We have incorporated and deleted sentences in lines 152-154.

8. Data need not be reported both in the text and in the Tables.

RESPONSE： Thank you for this suggestion. We have incorporated and the description regarding the mean values in the Results section has been removed.

9. Lines 180-182: why are these included in the subheading about physical activity time? 

RESPONSE： Thank you for this suggestion. We have incorporated and the relevant subheading has been removed as its deletion poses no issue.

10. Line 204: I believe the table lists an r=0.234 not r=0.237 as written.

RESPONSE： Thank you for this suggestion. Due to an error by the author, the value was 0.237; it has been changed to 0.234.

11. Line 299: restructure the sentence “… but with low predictive accuracy was low.”

RESPONSE： Thank you for this suggestion. We agree with you and have incorporated and Changed "accuracy" to "R2 values". This suggestion has been changed throughout our paper.

ｰｰｰｰｰｰｰｰｰｰｰｰｰｰｰｰｰｰｰｰｰｰｰｰｰｰｰｰｰｰｰｰｰｰｰｰｰｰｰｰｰｰｰｰｰｰｰｰｰｰｰｰｰｰｰｰｰｰｰｰｰｰｰｰｰｰｰｰｰｰｰｰｰｰｰｰｰ

Reviewer #2: Thank you for your scientific contribution. I have some concerns which need to be adressed:

RESPONSE： Thank you for dedicating your time and effort to reviewing my paper. Your constructive feedback was incredibly helpful in improving the manuscript. I appreciate your thoughtful comments, which strengthened the research.

1) Did you preregister the study? For example on OSF? Furthermore, no specific guidelines were followed such as STROBE for example.

RESPONSE： Thank you for this suggestion. You have raised an important point; however, we do not preregister this study as we believe it is a survey study and outside the scope of our paper.

2) Line 143-144:...and items that were found to be related in were used as independent variables. Which studies do you mean?

RESPONSE： Thank you for this suggestion. We have reflected on your comment and added the previous study's number in our paper (line# 165).

3) Were independence, variance homogeneity and normal distribution of residues tested for multiple regression?

RESPONSE： Thank you for providing these insights. Since no peculiar pattern was found in the plot of residuals, we assumed that the assumptions of independence and variance homogeneity were satisfied. The normality of the residuals was not mentioned in the previous manuscript, so it was added. (line #169).

4) Line 153-155: Too many word repetitions

RESPONSE： Thank you for this suggestion. We have incorporated and deleted sentences in lines 152-154.

5) Line 160-171: Should be in the methods section

RESPONSE：Thank you for this suggestion. We have incorporated and filled in lines 160-171 has been inserted into the procedures section of the method. (lines #137-#150)

6) Line 171: "Table 2 presents the PA times on holidays and their percentages." I cannot find any percentages

RESPONSE： We have incorporated your comments in Table 2. Sedentary time was calculated as a percentage by the number of minutes awake (1440 minutes minus average sleep hours).

7) Line 174-179: Should be integrated in discussion section

RESPONSE： Thank you for your suggestion. We have incorporated your suggestion and added the text to the Physical Activity Time section of the Discussion. (lines #234-#240)

8) Line 174: [19] appears to be cited wrongly

RESPONSE： Thank you for this suggestion. We have reflected your comments and revised the cited references in our paper (line #234).

9) Line 182: Did you mean 0.05 istead of 0.5? You did already state that in the methods section. RESPONSE： Thank you for pointing this out. We agree with you and have deleted line 182.

10) Line 298-299: Reword this sentence

RESPONSE：Thank you for pointing this out. We have reorganized the Conclusions section based on input from you and other reviewers. We think these changes now better. We hope that you agree.

11) Line 301: Use "pandemic" instead of "epidemic"

RESPONSE： Thank you for this suggestion. We have reflected your comments and the "pandemic" in our paper (excluding the questionnaire and the cited reference) and in the sections you mentioned

Reviewer #3: I thank the authors for sharing their work “Relationship between physical activity and neighborhood environment in preschool children during COVID-19.” I appreciate the opportunity to be able to review their work and would like to provide suggestions for major revisions. I have provided detailed comments below. I have noted several grammatical, punctuation, and formatting concerns in the manuscript and tables – correcting these would enhance the overall quality of the manuscript. Throughout the manuscript, some sentences remain fragmented so it is challenging to infer logical connections.

I hope that my feedback is helpful and would be happy to provide additional clarification as needed.

RESPONSE： Thank you for dedicating your time and effort to reviewing my paper. Your accurate and constructive feedback was very helpful in improving the manuscript. Your thoughtful comments are appreciated.

Abstract

- Line 26 – please consider elaborating on what is meant by “optimal health”

 RESPONSE： Thank you for this suggestion. In checking the body of our paper, we thought that it would be more appropriate to refer to "optimum developmental " rather than "optimum health. We added about optimal development. (line #26)

- Line 29 – does the term “holidays” refer to weekends (Saturday and/or Sunday) in the context of this paper? If yes, I would suggest potentially changing “holidays” to “weekend” throughout your manuscript.

RESPONSE： Thank you for this suggestion. in the body of our paper to holiday in the text has been changed to "weekend day".

- Line 32 – please consider starting your sentence with “Physical activity” instead of the acronym “PA.”

RESPONSE： Thank you for this suggestion. We agree with you and have incorporated this suggestion throughout the abstract of our paper.

Introduction

- Line 49 – the authors refer to the coronavirus as “exerting a profound impact on human health.” Can you please elaborate on this to more specifically state what aspects of human health, wellbeing, mental health, etc. were impacted by coronavirus? I think this sentence could be strengthened with the additions of a citation.

RESPONSE： Thank you for providing these insights. The human health effects of COVID-19 infection transmission control measures are described in our paper, with emphasis on the reduction in physical activity. We removed "exerting a profound impact on human health." and hope that the deletion clarifies the points we attempted to make.

- Lines 55-56 – can you please clarify if these trends were observed in adults or in children and adolescents?

RESPONSE： Thank you for this suggestion. We have reflected this comment (line #57).

- Lines 62-66 – I appreciate the authors referencing a national physical fitness and exercise survey, however, these sentences could be improved for greater clarity. What deteriorated and what population (e.g., adults, children, adolescents) is this referring to?

RESPONSE： Thank you for this suggestion. We have rewritten (lines #63-#68) to be more in line with your comments. We hope that the edited section clarifies.

- Line 67 – I would suggest including a citation here.

RESPONSE： Thank you for this suggestion. We have already included references but moved the reference numbers to the end of the sentence because they may not be clear.

- Line 71 – could you please clarify if the guidelines are referring to physical activity guideline or exercise guidelines?

RESPONSE： Thank you for this suggestion. The reference was the Physical Activity Guidelines. The name of the guideline is included in the text. (lines #74-#75)

- Lines 78-79 – did you operationally define “neighborhood environment” in the context of your study? Could you please revise this sentence to clarify what is meant by “the extent and nature of both PA and sedentary behavior”?

RESPONSE： Thank you for this suggestion. We described and clarified in our paper the neighborhood environment in terms of the built environment of the neighborhood. (lines #78-#83)

- Lines 87-88 – you provide background on the pandemic and its impact on physical activity, sedentary time, etc., however, there is no mention of the COVID-19 pandemic in your aim. If this is pivotal to your work and since it is mentioned in the article title, I would suggest revising your aim to include mention of the pandemic.

RESPONSE： Thank you for this suggestion. We have reflected this comment by the purpose of introduction (lines #88-#91).

- I would suggest included a greater emphasis on the importance of including preschool aged children in research studies. You mention that this group is considered a vulnerable population earlier in the introduction – providing another sentence or two on the importance/significance of this work among preschoolers may help strengthen the introduction.

RESPONSE： Thank you for this suggestion. We agree with you and have incorporated this suggestion throughout our paper. (lines #92-#96)

Methods

- Were parents formally trained on how to use the questionnaire or how to observe their children? If so, it would be helpful to describe and would enhance the quality of the methods.

RESPONSE： That is an interesting query. To prevent infection in preschoolers, the facility refused to meet with us in person to explain the situation. To address this, we accepted questions by phone or email as appropriate.

- Is there any additional information available on the family of the subjects? For example, was it a two-parent hous

---

## [Editor Report · Decision Letter 1]

21 May 2024

Relationship between physical activity and neighborhood environment in preschool children during COVID-19.

 - A cross-sectional study using 24-hour activity records-

PONE-D-23-22742R1

Dear Dr. Yamaguchi,

We’re pleased to inform you that your manuscript has been judged scientifically suitable for publication and will be formally accepted for publication once it meets all outstanding technical requirements.

Kind regards,

Patrick WC Lau

Academic Editor

PLOS ONE

Additional Editor Comments (optional):

The revision can address all comments and suggestions from the previous reviewers.
---

## [Editor Report · Acceptance letter]

1 Jul 2024

PONE-D-23-22742R1 

PLOS ONE

Dear Dr. Yamaguchi, 

I'm pleased to inform you that your manuscript has been deemed suitable for publication in PLOS ONE. Congratulations! Your manuscript is now being handed over to our production team.

Kind regards, 

on behalf of

Dr. Patrick WC Lau 

Academic Editor

PLOS ONE